# Dynamics-Based Regulatory Switches of Type II Antitoxins: Insights into New Antimicrobial Discovery

**DOI:** 10.3390/antibiotics12040637

**Published:** 2023-03-23

**Authors:** Ki-Young Lee, Bong-Jin Lee

**Affiliations:** 1College of Pharmacy and Institute of Pharmaceutical Sciences, CHA University, Pocheon-si 11160, Republic of Korea; 2College of Pharmacy, Seoul National University, Seoul 08826, Republic of Korea

**Keywords:** type II antitoxins, intrinsically disordered region, dynamic allostery, toxin activation, new antibiotic discovery

## Abstract

Type II toxin-antitoxin (TA) modules are prevalent in prokaryotes and are involved in cell maintenance and survival under harsh environmental conditions, including nutrient deficiency, antibiotic treatment, and human immune responses. Typically, the type II TA system consists of two protein components: a toxin that inhibits an essential cellular process and an antitoxin that neutralizes its toxicity. Antitoxins of type II TA modules typically contain the structured DNA-binding domain responsible for TA transcription repression and an intrinsically disordered region (IDR) at the C-terminus that directly binds to and neutralizes the toxin. Recently accumulated data have suggested that the antitoxin’s IDRs exhibit variable degrees of preexisting helical conformations that stabilize upon binding to the corresponding toxin or operator DNA and function as a central hub in regulatory protein interaction networks of the type II TA system. However, the biological and pathogenic functions of the antitoxin’s IDRs have not been well discussed compared with those of IDRs from the eukaryotic proteome. Here, we focus on the current state of knowledge about the versatile roles of IDRs of type II antitoxins in TA regulation and provide insights into the discovery of new antibiotic candidates that induce toxin activation/reactivation and cell death by modulating the regulatory dynamics or allostery of the antitoxin.

## 1. Introduction

Toxin-antitoxin (TA) systems are ubiquitously found in the genomes of prokaryotes and constitute a tightly regulated network involved in cell maintenance and survival under environmental stresses such as antibiotic treatment, nutrient deficiency, and immune system response [1,2,3,4]. Notably, numerous pathogenic bacteria carry multiple copies of TA operons, and thus, this system has received increasing attention as a novel antibiotic cellular target [5,6,7]. For instance, *Mycobacterium tuberculosis*, the causative agent of human tuberculosis, contains almost 80 TA systems [8,9]. Among eight different types of TA systems, type II TA systems are the most prevalent and are fairly well characterized in terms of structures, biological functions, and feedback mechanisms [1,10]. Type II TA operons typically encode two small cytoplasmic proteins—a stable toxin and a rapid-turnover labile antitoxin—both of which are <150 amino acids in length. The toxins inhibit essential cellular processes, such as protein synthesis and DNA replication, and retard cellular growth. The cognate antitoxins counteract these functions by forming tight noncovalent interactions and/or sterically protecting the active site of toxins under normal growth conditions.

Most type II antitoxins act as a modular protein that consists of two spatially separated functional domains: (i) a well-defined N-terminal DNA-binding domain responsible for the repression of the cognate TA transcription and (ii) an intrinsically disordered region (IDR) at the C-terminus that undergoes folding upon binding to the structured toxin. DNA-binding domains are evolutionally exchangeable across antitoxins [11], producing many possible combinations of the DNA binding domain and its adjacent toxin-neutralizing IDR.

## 2. Dynamics-Based Regulatory Switches of Type II Antitoxins

There is growing evidence that IDRs from eukaryotes act as regulatory recognition elements that have high propensities for the secondary structure and form multiple layers of structure and protein–protein interaction (PPI) [12,13,14,15,16]. Notably, these IDRs are enriched in signaling-associated proteins (e.g., c-Myc and p53) or aggregation-prone proteins (e.g., amyloid β-peptide (Aβ) and α-synuclein), and their aberrant functions or altered abundance in cells can lead to cancer development or neurodegenerative disease [12,17]. The most important feature of functional IDRs is that they respond to stimuli (e.g., post-translational modifications, partner bindings, and pH changes) and provide varying levels of structural plasticity and heterogeneity for interactions with multiple partners to function. Targeting IDRs of disease-relevant proteins in cells is a new trending strategy in drug discovery [18,19,20]. It has been challenging to elucidate the structural details of the dynamics and heterogeneity of IDRs using conventional biophysical techniques; however, IDRs involved in various diseases are attractive therapeutic targets.

As IDRs are less prevalent in prokaryotes than in eukaryotes, the physicochemical property and functionality of IDRs in the prokaryotic proteome remain largely unknown. A body of experimental evidence has recently demonstrated that C-terminal IDRs of prokaryotic type II antitoxins have conditional dynamics-based modes of regulation of the TA system [4,11,21,22,23,24,25]. These findings have provided new insights into the functional diversity of prokaryotic IDRs. The degree of dynamics or disorder within antitoxin IDRs serves to generate versatile functions of antitoxins as a toxin binder and a repressor or derepressor of TA expression.

### 2.1. Coupling between Binding and Folding of Antitoxin IDRs

The folding-upon-binding process has been considered to inevitably lead to medium-to-low affinity interactions due to the energetic cost of folding [26]; however, it is typically not the case for TA pairs that have evolved to associate with a dissociation constant of the pico-to-nanomolar range [1,2]. The high affinity could be attributed to the combination of favorable intramolecular interactions of toxin-neutralizing IDRs, even in toxin-free states, and structural and electrostatic complementarities at the TA contact interface [11,27]. These characteristics may enable the antitoxin to overcome the large entropic cost of the binding-coupled folding. In addition, toxin neutralization induced by antitoxin binding has been proposed to employ conformational selection where only transiently preexisting folds within the structural ensemble are able to bind to the partner, and/or the induced fit mechanism in which all structures within the ensemble are captured by the partner and subsequently fold [11,28]. Both mechanisms promote the folded or ordered state of IDRs to favor binding to toxins. In agreement with these mechanisms, the crystal structures of TA complexes have revealed that antitoxin IDRs form α-helical states that bind to the corresponding toxin; however, it has been challenged that the IDRs are crystallized in the absence of the toxin partner [29,30,31].

### 2.2. Regulation of TA Transcription by Antitoxin IDR

Recent literature has revealed that the dynamics-based allostery of antitoxin IDRs is crucial for the autoregulation of type II TA operons, termed “conditional cooperativity” [2,4,22,25]. This mechanism could explain the effects of the relative stoichiometry between toxin and antitoxin molecules at the level of cognate transcriptional repression, as demonstrated by type II TA subclasses including CcdAB, Phd-Doc, RelBE, VapBC, ParDE, and Kid-Kis [25]. The molecular basis lies in how the toxin level allosterically influences affinity for the interaction between antitoxins and two closely located operators of palindromic sequences in its own promoter region.

Antitoxins act as transcriptional repressors to associate with one or two neighboring DNAs on the TA operon; the antitoxin–DNA association is stabilized by the positive cooperativity mechanism, where one cognate toxin is an allosteric link that bridges two C-terminal IDRs of DNA-bound antitoxins. When antitoxin IDRs are fully occupied by toxins, steric repulsion between adjacent toxins occurs, which destabilizes the repressor complex. It should be noted that this molecular process can employ the disorder-to-order transition of antitoxin IDRs as structural factor. For instance, to achieve the toxin-mediated stable complex on operators, the IDR of the antitoxin Phd requires two modes of TA interaction with two different affinities for non-overlapping binding sites of the Doc toxin [22,32]; the first high-affinity binding leads to an α-helical conformation within the IDR more readily than the second low-affinity binding that may mediate the formation of the “fuzzy” TA complex (Figure 1A).

Notably, the low-to-high affinity transition has been thought of as a key step in TA operon regulation [23,33]. The switch of binding enables formation of the TA complex with a 1:1 ratio; this generates steric hindrance and destabilizes the repressor complex previously driven by the low-affinity binding, resulting in release of the antitoxin from the operator. On the other hand, IDRs of toxin-free Phd are involved in negative cooperativity in TA transcription by providing an entropic repulsion or conformational restrictions to prevent a second Phd molecule from binding to the adjacent operator (Figure 1A) [23]. Doc can interact with the IDR of Phd and induce the folding of the IDR to break down the entropic barrier, resulting in the switch from negative to positive cooperativity in transcription repression at high Doc-to-Phd ratios. Such additional regulation involving the entropic repulsion has not been found in the CcdAB system, where the affinity between free antitoxin and operator is inherently very weak [34,35].

The mechanism involving the low-to-high affinity switch is not the case for the *RelBE* TA system from *E. coli* (Figure 1B) [36,37]. Two RelB dimers associate cooperatively with adjacent operator sites even in the absence of the RelE toxin, and the RelB-DNA complex is further stabilized by binding of the RelE monomer to the RelB dimer as a 1:2 complex. The bound RelE may promote a folded, DNA-binding competent state of RelB. Excess of RelE will promote the TA interaction, which outcompetes the contacts between the adjacent RelB dimers and generates steric hindrance that destabilizes the TA pair-operator complex and activates transcription. More recently, the same mechanism was found in the YoeB-YefM system from *Staphylococcus aureus,* which exhibited that with a 1:1 ratio of antitoxin and toxin, two YoeB molecules disassembles the heterohexamer (YoeB-YefM_2_-YefM_2_-YoeB) with a high DNA-binding affinity to two heterotetramers (YoeB-YefM_2_-YoeB) incapable of DNA binding due to steric clashes [38].

Similar but more complex mechanisms have been observed for GCN5-related *N*-acetyl-transferase (GNAT)-toxin-antitoxin modules, such as *E. coli ataRT* (Figure 1C) [39,40] and *Klebsiella pneumoniae* [41]. The AtaR antitoxin binds to the AtaT toxin in an inactive monomeric state and neutralizes the toxin that catalyzes acetylation of Met-transfer RNA^fMet^ and lead to growth arrest. AtaR IDRs can interact with AtaT at two different sites and folds into different structures, which enables the IDR to have dual functions: toxin neutralization and positioning the DNA-binding domains of AtaR in an orientation that promotes formation of the AtaRT-DNA repressor complex. Excess of AtaT can result in a TA complex with a 2:2 stoichiometry that causes a steric clash between the complexes, dissociates from the operators, and de-represses transcription. However, it remains largely unknown how type II antitoxins on two adjacent operator DNAs are allosterically linked and conformationally altered by the toxin bound to the IDRs. In a few TA modules, such as *mqsRA* from *E. coli* and *hicAB* and *graTA* from *P. putida* [42,43,44,45], the toxin acts as a transcription derepressor to reduce the affinity between its operator and antitoxin, which lacks the IDR and is fully structured, in contrast to the IDR-mediated cooperative TA binding to their operators. For instance, unlike many other type II antitoxins, the MqsA antitoxin contains two partially overlapping folded domains, the operator-binding and MqsR-neutralizing domains. Thus, MqsR binding sterically inhibits the operator binding of MqsA, resulting in de-repression of the *mqsRA* operon [43].

Several type II TA systems possess an inverse organization in which the toxin-encoding gene precedes the gene encoding the antitoxin [46]. These systems employ a new regulatory mechanism to secure selective synthesis of antitoxin in the presence of excess toxin; this allows only antitoxin expression and homeostatic maintenance of a low toxin-to-antitoxin ratio. For instance, *mqsRA* promoters located in the preceding sequence for the MqsR toxin drive the constitutive production of the MqsA antitoxin [47]. There are type II TA operons that contain a single operator recognized by the antitoxin dimer, and the antitoxin and the cognate TA complex bind to the operator with a similar affinity, as exemplified by the *E. coli higBA* module; however, HigA undergoes a large conformational change upon binding to HigB [48]. As with the *higBA* module, the *E. coli dinJ-yafQ* system exhibited that both DinJ and the DinJ-YafQ complex repress the transcription to a similar extent [49]. A notable difference is that the *dinJ-yafQ* module is controlled externally by the transcription repressor LexA, which is activated through the SOS response system. However, if antitoxins allow allosteric communication between the toxin- and DNA-binding sites, we could hypothesize a different type of conditional cooperativity by which the allosteric effects of one or two toxins on binding of the antitoxin to the single operator differ significantly, even though the two sites appear to be spatially distinguishable. Such interdomain communication of the antitoxin has been suggested by the Phd-Doc system in which Doc binding increases the propensity for helical formation of the Phd IDR, which then propagates to the DNA-binding domain (Figure 2), resulting in stronger DNA binding [22].

The propagation between the distant binding sites can be interpreted as a relay of change in protein dynamics and flexibility, known as dynamic allostery [50], or within defined sets of visible conformations. We believe that the presence of multiple antitoxin-binding sites on the operator adds multiple levels of the complexity to the cooperative repression/de-repression mechanism for TA transcription. In addition, it is possible that, if promiscuous and different modes of interaction between antitoxin and paralogous toxins are present in the same organism, these associations would reduce the number of cognate TA complexes and attenuate the transcription repression.

### 2.3. Auto-Inhibitory States of Antitoxin

C-terminal IDRs of toxin-free antitoxins may fluctuate between distinct conformational states and allosterically regulate the function of antitoxins. For instance, the IDR of the CcdA antitoxin exhibited multiple sets of peaks on the ^15^N–^1^H heteronuclear single quantum coherence (HSQC) spectra, demonstrating that this disordered region adopts several distinct conformations in equilibrium on the NMR time scale [51]. Single peaks for a given nucleus may represent an average of the resonance for fast-exchange dynamic conformations. Computational simulations coupled with NMR observables and single-pair Förster resonance energy (spFRET) data have suggested that CcdA coexists primarily in three representative conformations, referred to as “closed”, “partially closed”, and “open” states, which involve two flexible C-terminal IDRs of antitoxins [21]. In the closed state, both IDRs contact the folded N-terminal domain and form a relatively compact structure (Figure 3).

This state should sterically prevent this domain from DNA binding and TA transcription repression and sample “inhibitory states” that are incompatible with toxin binding. In other words, transient intramolecular interaction of disordered and folded regions of the protein can competitively inhibit the binding of the ligand with the folded domain. By contrast, the partially closed state shows that only one of the two C-terminal IDRs associates with the N-terminal domain, and the open state adopts an ensemble of IDR extended conformations similar to random coils (Figure 3). The open state should be more accessible to either proteases or toxins than the closed state; thus, it is conceivable that the increased number of open states promotes not only protease-mediated degradation but also toxin neutralization. The balance between these molecular events remains to be elucidated in a physiological context.

Based on these observations, we speculate that the equilibrium distribution between the three states can alter under environmental stresses and that the population shift has substantial influence on the regulatory network in TA systems. It should be considered that differentially populated dynamic states in the IDR ensemble can be influenced by the extremely crowded cellular environment, that provides local high concentrations of biological macromolecules, such as carbohydrates, nucleic acids, and proteins [52,53]. These crowding molecules could be inert or active components that form direct physical contacts with IDRs, potentially promoting either folding or unfolding of the IDR in a sequence-dependent manner. In particular, the crowded environment may cause considerable restrictions on the amount of free water and thus decreases volume availability of biomolecules, which can affect the conformational equilibrium of IDRs [54]. However, the crowding effect on IDRs of type II antitoxins remains unknown; whenstudying this conformational dynamics, in vitro biophysical techniques (e.g., NMR, SAXS, DLS) will be applicable in the presence of biologically inert crowders such as polyethylene glycol, Ficoll, and Dextran polymers [55,56,57]. Dynamics-based allostery of antitoxin IDRs has been accepted as a new paradigm beyond the simple mechanism by which binding and folding are coupled to inhibit the toxin’s activity, providing additional insights into the regulatory mechanism for cell growth and survival [11,21,22,23].

### 2.4. Proteolysis of Antitoxin IDRs by Proteases

The activity of antitoxin IDRs commonly has a short in vivo lifetime since these disordered segments are thermodynamically unstable and susceptible to degradation by intracellular proteases. For instance, it has been shown that activation of the Lon protease leads to the proteolysis of type II antitoxins, the release of toxins, and translation arrest in the context of bacterial persistence [58] and that plasmid-encoded antitoxins (CcdA, Kis, and Phd) are constitutively degraded by cellular proteases (Lon, ClpAP, and ClpXP, respectively), supposedly allowing the liberation of toxins in plasmid-free segregants [59,60,61].

Although detailed molecular mechanisms for the antitoxin degradation process on a case-by-case basis and its substrate specificities for proteases are not well understood, it has been recognized that antitoxin IDRs are more sensitive to proteolysis than their cognate toxins and thus that rapid proteolysis of the antitoxin is a critical step to liberate and activate the toxin; however, this notion has recently been challenged [62,63]. The extended and disordered conformation of the IDR may expose recognition sites of cellular proteases to the solvent. However, the lack of conservation in the antitoxin IDR sequences has made it challenging to predict how TA systems interact with cellular proteases and how much this interaction influences cycling between the activation and inactivation states of toxins [2]. It is unknown whether an antitoxin within the TA complex can be directly targeted by proteases, although it appears that the antitoxin is protected from degradation once a stable TA complex is formed [62]. It still remains possible that, in the TA complex, the disordered C-terminal free ends of the antitoxin are accessible to cytoplasmic proteases and that the formation of less stable non-cognate TA complexes leads to increased sensitivity to proteases [64]; this potentially reduces the stability of the repressor complex on the operators. In bacteria, it has been suggested that type II antitoxins are degraded by multi-subunit machines such as ATP-dependent proteases Lon or Clp, which reconstitute the cellular proteome under certain physiological conditions including environmental adaptation [65], altered levels of misfolded proteins [66], and temperature changes [67]. The proteolysis of antitoxin IDRs should increase the de-repression of the TA operon transcription, and the subsequent replenishment of antitoxins inhibits reactivation of toxins.

## 3. Utilization of Antitoxin IDR Dynamics for New Antibiotic Discovery

Over the past decades, worldwide distribution of antibiotic-resistance pathogens has emerged as a modern health issue, and considerable efforts have been made to develop new classes of antimicrobial agents. Type II TA systems are most extensively studied and occur frequently in human pathogenic bacteria. It has been widely accepted that the activation of type II TA system initiates the coexpression of cognate toxin and antitoxin proteins from the same operon, followed by formation of the TA complex from which toxin can be liberated and active due to the increased degradation of antitoxin relative to synthesis [68,69]. However, antibiotic developments utilizing this system are still lacking and remain challenging since primary biological roles of type II TA systems have been strongly questioned and appear contradictory; the TA activation would result in two apparently opposite outcomes: induction of cell death and survival under stress condition. For instance, activation of the MazF toxin can lead to cell death or the reversible inhibition of cell growth, which is required for the generation of persister cells that are able to survive stress conditions [70,71,72]. However, the role of the toxin in cell survival conflicts with the recent data that the toxin is not freed to function during stress, although the stress induces the increased transcription level of chromosomally encoded type II TA systems in *E. coli* [62]. This observation agrees with the frequently raised possibility that the TA system is not a critical effector of bacterial persistence. However, this does not necessarily imply that the chromosomal TA system can induce cell death as observed in the absence of plasmid-encoded TA modules [1,2,73]. Despite these unresolved issues, one would conceive that cell death can be largely induced by the hyperactivation of toxin, thereby altering the balance between death and survival.

The toxin-neutralizing IDRs of type II antitoxins described here act as a “hub” component of the TA system regulation; as such, the IDRs could be considered as a promising antibiotic target. However, despite their biological and therapeutic importance, it has been challenging to elucidate the details of the functional dynamics or allostery of the IDR alone or in complex with potential ligands. In aid of computational simulations, recent progress has provided insights into the potential for targeting IDRs via dynamic and nonspecific interactions of small molecules with multiple sites to form “dynamic complexes” [18,20,74,75,76,77]; this is in contrast to the traditional paradigm in which drugs bind to the protein targets with specific conformation. It would be envisaged that ligand binding may modulate the structural ensemble of IDR by (i) promoting one of ensemble conformational states with an overall reduction of entropy (i.e., entropic collapse), (ii) redistributing the states without overall entropy change (i.e., isentropic shift), or (iii) increasing the number of the states with an overall entropic increase (i.e., entropic expansion) [78].

Approaches utilizing such ensemble modulation mechanisms are applicable to targeting IDRs of many type II antitoxins. It is noteworthy that the intrinsic property of IDRs that preexist in the ensemble of numerous conformations is implicated in generating more cavities for ligand binding than well-folded proteins [19], and the improvement of molecular dynamics (MD) simulation methods for these flexible cavities in the ensemble should contribute to judging the druggability of IDRs. One would optimize ensemble docking protocols using the number of IDR conformers, transition rates and pathways between conformations, and IDR force fields. Subsequently, MD snapshots for transiently formed druggable cavities will be exploited for structure-based virtual screening to identify potential drug candidates. Combined with computation-based methods, in-solution biophysical techniques (e.g., NMR, SAXS, and single-molecule spectroscopy) can be used for the screening of potential ligands that directly bind to druggable cavities, as well as constructing nonfunctional conformation ensembles or interfacial binding modes of the IDR–ligand complexes at the atomic resolution. Covalent binding of the ligand to reactive residues of the IDR would be another potential strategy for the inhibition of active conformation at low doses and for a long time.

Alternatives to small molecules, larger molecules, peptides (10–50 amino acids), proteins (>50 amino acids), and peptidomimetics can be engineered to target IDRs. Notably, these inhibitors have been recognized to competitively disrupt PPIs due to their intrinsic abilities to bind to relatively large and shallow protein surfaces. However, linear peptides are difficult to pass through cell membranes and are inherently unstable since they undergo proteolysis in cells. To tackle these challenges, chemical modifications of peptides by intramolecular stapling or cyclization have recently emerged as powerful techniques to increase stability, binding affinity, selectivity, and cell permeability of peptides [79,80]. Both techniques can produce conformationally constrained analogs and reduce the entropic cost of binding to IDR targets compared with linear peptides. For instance, cyclic peptide inhibitors have recently been developed to target the disordered region of neutrophil extracellular traps (NET)-resident histone H2A that induces the reduction in atheroprogression [81]. In addition, a larger surface contact of IDR with peptides rather than small compounds may facilitate less dynamic but more stable complexes.

Understanding the structural aspects of cooperative interactions between antitoxin, toxin, and their operator DNA involves dissecting the mechanism for the conditional repression of TA transcription and its coupling with toxin neutralization, as well as design of potential antibiotics that activate endogenous toxins and induce cell death. Due to the ubiquitous distribution of type II TA modules in human pathogenic bacteria, they have been recognized as promising antibiotic targets. With time and in efforts of developing novel antibiotic strategies targeting type II TA systems, some critical questions have often been asked about consequences of the TA activation: for instance, how many free toxins are enough to induce cell death? Does the TA transcription necessarily need to be suppressed to reduce supplemental antitoxins and increase the amounts of free toxins to exert their bactericidal effects? Does the toxin-induced cell death depend on subtypes of type II TA systems? Antibiotics targeting type II TA systems must be developed to fully eradicate relevant pathogenic cells and prevent recurrence, which will potentially evade bacteriostasis induced by the formation of persister cells and biofilm. It should also be considered that these systems have evolved to differ in structure, function, and regulatory mechanism and to have distinct modes of action despite their structural similarities [1,4,82]. Type II TA modules are generally not conserved even in closely related bacteria. In addition, type II antitoxins have been comprised of the DNA binding domain and different families of the toxin-neutralization domain, enabling different modes of bridging the operator and toxin within the TA-DNA transcription repressor. This dynamic evolution and its limited information have, so far, made it challenging to establish the general rule for antibiotic strategies targeting type II TA systems. However, despite this challenge, we believe that antitoxin IDRs are promising targets of therapeutic value since they act as key components to mediate both toxin inhibition and autoregulation of the TA operon.

Based on the dynamics-based multiple roles of the antitoxin IDR in the TA system, several therapeutic strategies can be constructed as follows (Figure 4).

First, the toxicity of the toxin is artificially increased by targeting antitoxin IDRs to prevent or disrupt non-functional TA complexes. Second, antitoxin-binding sites of the toxin are sterically blocked by IDR-mimicking agents while retaining substrate accessibility to the toxin, which has been exemplified by PPI inhibitors of PemIK, MoxXT, and VapBC complexes [83,84,85]. Third, the toxin-enhanced complex between antitoxin and operator DNA is promoted by “indirect” transcription modulators that stabilize the toxin bridge that connects two IDRs of the DNA-bound antitoxins. The cellular effect of these modulators may arise from inhibited cotranscription of the cognate toxin and antitoxin, which resembles those of TA gene *loci* deletion. This is reminiscent of the initially identified role of TA genes (e.g., F-plasmid *ccdAB*) within bacterial plasmids in post-segregational cell killing (PSK) to maintain a bacterial population with these replicons [86]. This maintenance is due to the elimination of plasmid-free daughter cells where antitoxins are more amenable to proteolysis than toxins, but not supplemented by the TA operon, facilitating the activity of free toxins. However, it remains controversial as to whether and which chromosomal TA modules are involved in cell death that is similar to PSK mediated by plasmid-encoded TA systems [62]. It is possible that such bactericidal effect is elicited by stabilizers of the TA pair-operator complex that represses TA transcription. If a system employs a low-to-high affinity transition to dissociate the TA complex from operators in the conditional cooperativity mechanism, described above, one could design allosteric modulators to promote the low-affinity TA interaction within the repressor complex while retaining the high affinity for toxin neutralization. It is still unclear how much type II toxins liberated in the absence of the TA transcription contribute to the induction of cell death, besides their reported biological roles in the formation of persister cells under certain stringent conditions [5,58]. However, the therapeutic design to promote formation of the TA repressor complex is not applicable to unusual “reversed” TA configurations with the toxin gene being upstream of the antitoxin gene since these systems do not employ the regulatory mechanism where toxin acts as a conditional corepressor of transcription. This strategy is also incompatible with infrequent three-component TA systems in which the antitoxin solely has the neutralizing function, and transcription is regulated by separate third component [87,88]. Fourth, IDR conformational states that are sensitive to cellular proteases are directly or allosterically stabilized. This strategy would be plausible since the regulatory feedback loop in type II TA systems typically ensure stoichiometric excess of antitoxins over cognate toxins in steady-state conditions. Even in the absence of this regulatory mechanism, antitoxins may be more abundant than the cognate toxin as the type II antitoxin gene in the TA operon is typically more efficiently translated than the downstream gene encoding the cognate toxin. Free antitoxins prior to the TA complex might adopt protease-sensitive conformations in the IDR. Thus, their artificial proteolysis by antibiotic candidates causes the lack of association between the IDR-truncated antitoxin and toxins liberated spontaneously from the preformed TA complex, resulting in activation of the liberated toxin. Recent experimental data support the premise of our strategy, where free antitoxin is preferentially degraded when compared to antitoxin in the TA complex [62]. Last, potential auto-inhibitory closed states of antitoxins, in which toxin-binding sites are inaccessible to the solvent, are stabilized by ligands that mediate intramolecular interactions between IDR and folded domain. This strategy should also be valid in the condition described in the fourth, which provides toxin-free antitoxins containing the IDR that dynamically forms a closed state incapable of toxin neutralization. It should be noted that the latter two strategies are not applicable to a few antitoxins in which the toxin-neutralizing domain is stably folded even in the absence of the cognate toxin [43,44,89,90].

To be more effective or applied to a broader spectrum of pathogenic bacteria, it would be advantageous to use appropriate combinations of TA-targeting agents and conventional antibiotics. With the advent of the era of using optimized computational and experimental methods for targeting disease-associated IDRs, type II antitoxin IDRs that function as a central hub in the regulation of the TA system hold high potential of new reliable targets for the treatment of antibiotic-resistant bacteria infections.

## Figures and Tables

**Figure 1 antibiotics-12-00637-f001:**
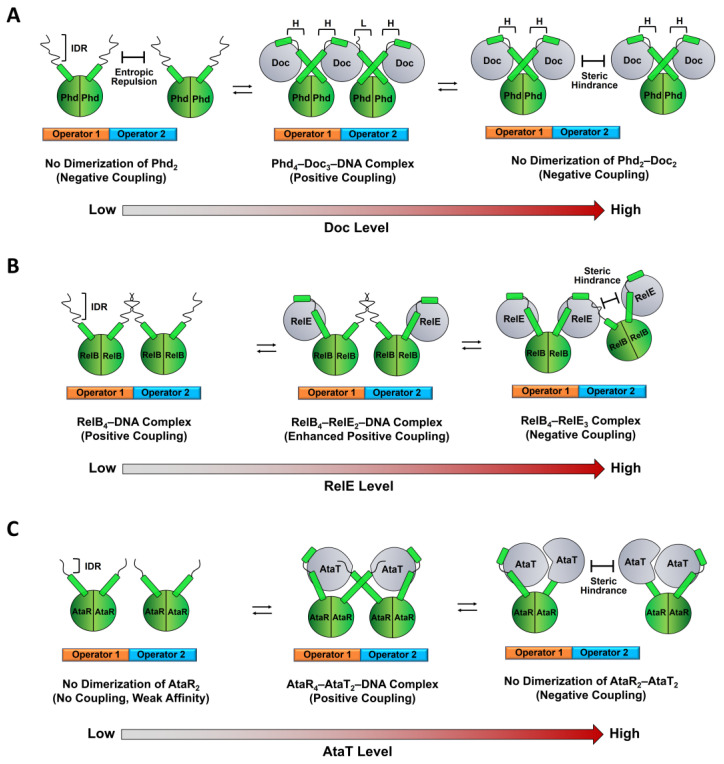
Schematic of the representative molecular mechanism for conditional cooperativity in the TA operon regulation. (**A**) phd-doc. The C-terminal IDR of Phd that associates with its own operator inhibits the binding of a second Phd dimer to the adjacent operator at low Doc-to-Phd ratios. This mechanism involves an entropic repulsion of the Doc-free IDR of Phd against another Phd IDR, leading to a negative cooperativity and weak transcriptional repression. As the level of Doc increases, binding of Doc induces the folding of the IDR to reduce the entropic repulsion and promotes a stable repressor complex, resulting in a switch to positive cooperativity. In this complex, two Phd dimers on operators are bridged and stabilized by one Doc molecule that binds to two Phd IDRs with two different affinities. “H” and “L” represent relatively high and low affinities, respectively. Addition of Doc will stochastically promote the low-to-high affinity transition and the formation of a Phd-Doc complex in which all IDRs of the Phd dimer bind to Doc. Steric repulsion between the bound Doc molecules can occur and preclude the formation of a stable repressor complex, resulting in de-repression of the *phd–doc* operon transcription. (**B**) *relBE*. The RelB dimer of dimers is cooperatively formed via their IDRs on two adjacent operators. This RelB-DNA complex is further stabilized by two RelE monomers that flank the RelB dimer. Addition of RelE generates steric hindrance between RelE molecules within the RelBE-operator complex, weakens the DNA binding of RelB, and activates transcription. (**C**) *ataRT*. The AtaR antitoxin binds to its operator DNA with a weak affinity. Formation of the AtaR-AtaT complex with a 2:1 stoichiometry promotes a favorable orientation in which the DNA-binding domain of AtaR associate with the operator with high affinity. Higher AtaT:AtaR ratios result in a TA complex with lower affinity for the operator DNA.

**Figure 2 antibiotics-12-00637-f002:**
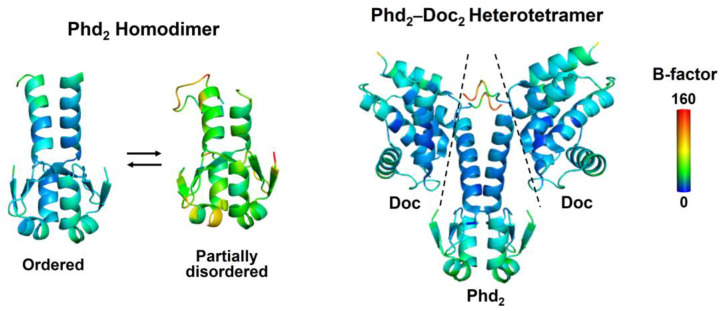
Crystal structures of the antitoxin Phd alone and in complex with the cognate toxin Doc. The antitoxin Phd exists in ordered or partially disordered conformations as observed in crystal structures [22] (accession number: 3HS2 for the ordered state and 3HRY for the partially ordered state). An ordered conformation of Phd in the complex with Doc is observed in the crystal structure [22] (accession number: 3K33). Crystallographic B-factors are color-mapped onto the structures. Dotted lines represent the interface between Phd and Doc.

**Figure 3 antibiotics-12-00637-f003:**
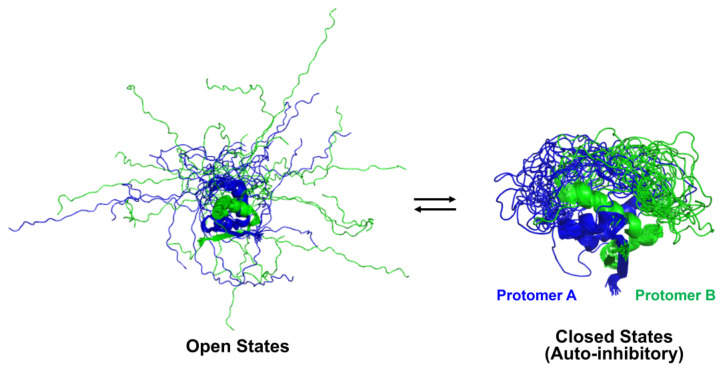
Open and closed states of the antitoxin CcdA in solution. Both open and closed states of the antitoxin CcdA in the absence of the cognate toxin CcdB are described as an overlay of the 20 best NMR structures in solution.

**Figure 4 antibiotics-12-00637-f004:**
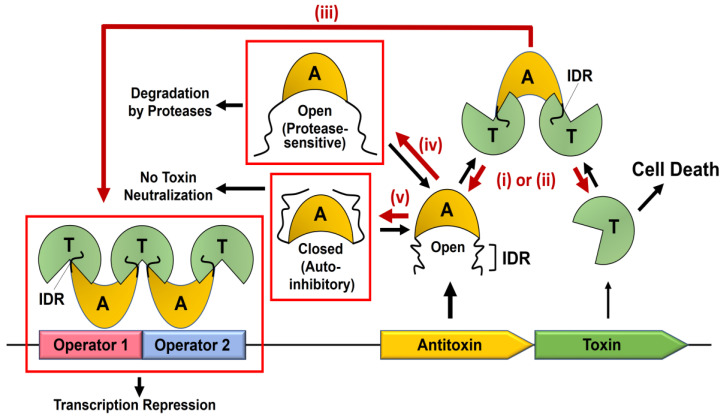
Proposed antimicrobial strategies utilizing the dynamics-based allostery of type II antitoxin IDRs. General mechanisms for toxin neutralization and transcription regulation in type II toxin-antitoxin systems are schematically described, and the molecular processes that can be promoted by potential antibiotics are represented by red arrows. These processes are labeled with five antibiotic strategies–(i), (ii), (iii), (iv), and (v)–as explained in the text.

## Data Availability

Not applicable.

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
