# Peer review of "Dynamics-Based Regulatory Switches of Type II Antitoxins: Insights into New Antimicrobial Discovery"

_antibiotics, 2023, doi:10.3390/antibiotics12040637_

Round 1

Reviewer 1 Report

In the manuscript “Dynamics-Based Regulatory Switches of Type II Antitoxins: Insights Into New Antimicrobial Discovery”, the authors described the current state of knowledge about the roles of IDRs of type II antitoxins in toxin-antitoxin regulation. They concluded the review with the possibility of designing new antibiotic candidates that modulate the regulatory dynamics or allostery of the type II antitoxin by targeting IDRs in order to induce toxin activation/reactivation and, thus, cell death. This review is pertinent for the type of manuscript, ‘Perspective’ but I have following comments.

Major comments:

1. References are missing in the review. I recommend the authors to describe in more detail the conditional cooperativity mechanisms of other TA systems like HicAB, RelBE, DinJ-YafQ, MqsRA...a figure with the different mechanisms will be interesting to have an overview of the literature instead of the Figure 1 with just the Phd-Doc example. Examples of references: PMID: 32845304, PMID: 31101334, PMID: 30337369, PMID: 28266056…

2. It is important to understand the interest of the design of antibiotics that target IDRs of type II TA system. The strategy will be different depending on each antitoxin. It would be interesting to provide strategies for some examples of antitoxins related to Figure 4.

3. The figure 3 is missing.

4. Figure 1: indicate the IDRs in the Figure.

5. Figure 4: indicate the IDRs in the TA complexes. The red arrows can be replaced by small molecules like cyclic peptides.

6. An example of molecular dynamic simulation to propose new antibiotics targeting IDRs of type II antitoxins will be really interesting to improve the manuscript.

Minor comments:

1. line 11: add ‘the type II TA system’ instead of ‘the TA system’ alone

2. line 52: remove ‘and function in physiological condition-dependent manners’

3. line 56: the term ‘undesirable results’ is not appropriated. Please change.

4. line 87: the term ‘reluctant’ is not appropriated. Please change.

5. line 94: ‘Kid-Kis’ instead of ‘Kis-Kid’.

Author Response

COMMENT 1> In the manuscript “Dynamics-Based Regulatory Switches of Type II Antitoxins: Insights Into New Antimicrobial Discovery”, the authors described the current state of knowledge about the roles of IDRs of type II antitoxins in toxin-antitoxin regulation. They concluded the review with the possibility of designing new antibiotic candidates that modulate the regulatory dynamics or allostery of the type II antitoxin by targeting IDRs in order to induce toxin activation/reactivation and, thus, cell death. This review is pertinent for the type of manuscript, ‘Perspective’ but I have following comments.

RESPONSE 1> We appreciate the Reviewer’s kind evaluation of our manuscript. We carefully addressed the Reviewer’s points in the revised manuscript. Overall, we improved our manuscript mainly by adding more description and discussion about the regulatory mechanisms of type II TA systems and the antibiotic strategies to the chapter 2.2 on page 2-5 and the chapter 3 on page 7-10, respectively.

COMMENT 2> References are missing in the review. I recommend the authors to describe in more detail the conditional cooperativity mechanisms of other TA systems like HicAB, RelBE, DinJ-YafQ, MqsRA...a figure with the different mechanisms will be interesting to have an overview of the literature instead of the Figure 1 with just the Phd-Doc example. Examples of references: PMID: 32845304, PMID: 31101334, PMID: 30337369, PMID: 28266056…

RESPONSE 2> As the Reviewer suggested, we have added the description of the conditional cooperativity mechanisms of other type II TA systems and cited the above and other related references in the chapter 2.2. “Regulation of TA transcription by antitoxin IDR” on pages 2-5 in the revised manuscript. Accordingly, we included relevant figures (B) and (C) in the Figure 1 on page 3.

COMMENT 3> It is important to understand the interest of the design of antibiotics that target IDRs of type II TA system. The strategy will be different depending on each antitoxin. It would be interesting to provide strategies for some examples of antitoxins related to Figure 4.

RESPONSE 3> We thank the Reviewer for raising this valuable point. As the Reviewers suggested, we have added discussions about antibiotic strategies targeting antitoxin IDRs, including the subtype-dependent antibiotic design, in the chapter 3 “Utilization of antitoxin IDR dynamics for new antibiotic discovery” on pages 7-10 in revised manuscript.

COMMENT 4> The figure 3 is missing.

RESPONSE 4> We confirmed that this figure is included on page 6 in the revised manuscript.

COMMENT 5> Figure 1: indicate the IDRs in the Figure.

RESPONSE 5> We indicated the IDRs in the Figure 1 on page 3 in the revised manuscript.

COMMENT 6> Figure 4: indicate the IDRs in the TA complexes. The red arrows can be replaced by small molecules like cyclic peptides.

RESPONSE 6> As the Reviewer suggested, we indicated the IDRs in the TA complexes in Figure 4 on page 9 in revised manuscript. In addition, we tried to replace the arrows with potential antibiotics like small molecules or peptides in the Figure 4, however, we noticed that this replacement looks more complicated and unclear. Thus, we did not add these molecules to the Figure 4.

COMMENT 7> An example of molecular dynamic simulation to propose new antibiotics targeting IDRs of type II antitoxins will be really interesting to improve the manuscript.

RESPONSE 7> We thank the Reviewer for this interesting point. However, molecular dynamics simulation data for type II antitoxin IDRs with potential drug-binding cavities or those in complex with antibiotic candidates have not been reported. Thus, we could not add a description of these molecular dynamics data to the revised manuscript.   

COMMENT 8> Minor comments:

  1. line 11: add ‘the type II TA system’ instead of ‘the TA system’ alone
  2. line 52: remove ‘and function in physiological condition-dependent manners’
  3. line 56: the term ‘undesirable results’ is not appropriated. Please change.
  4. line 87: the term ‘reluctant’ is not appropriated. Please change.
  5. line 94: ‘Kid-Kis’ instead of ‘Kis-Kid’.

RESPONSE 8> As the Reviewer commented, we removed or corrected inappropriate expressions in the revised manuscript.  

Reviewer 2 Report

In this review, Fraikin et al. give a comprehensive understanding of type II toxin-antitoxin systems in bacteria. The review provides an in-depth analysis of the evolution and functions of these systems, including their role in bacterial stress response, persistence, and antibiotic resistance. The review also covers the recent advances in our understanding of these systems, such as the discovery of new toxin-antitoxin modules and their mechanism of action. Overall, this review is well-written, and informative, and offers a comprehensive and up-to-date overview of the current state of knowledge on type II toxin-antitoxin systems.

Author Response

COMMENT 1> In this review, Fraikin et al. give a comprehensive understanding of type II toxin-antitoxin systems in bacteria. The review provides an in-depth analysis of the evolution and functions of these systems, including their role in bacterial stress response, persistence, and antibiotic resistance. The review also covers the recent advances in our understanding of these systems, such as the discovery of new toxin-antitoxin modules and their mechanism of action. Overall, this review is well-written, and informative, and offers a comprehensive and up-to-date overview of the current state of knowledge on type II toxin-antitoxin systems.

RESPONSE 1> We appreciate the Reviewer’s overall positive evaluation of our manuscript. As the Reviewers suggested, we improved our manuscript mainly by adding more description and discussion about the regulatory mechanisms of type II TA systems and the antibiotic strategies to the chapter 2.2 on page 2-5 and the chapter 3 on page 7-10, respectively.

Reviewer 3 Report

This is with reference to the review title “Dynamics-Based Regulatory Switches of Type II Antitoxins: Insights into New Antimicrobial Discovery by Ki-Young Lee and Bong-Jin Lee. They have discussed about the dynamic-based regulatory switched of type II antitoxin with specific reference to Phd-Doc/ CcdAB system having two operators. I have some major concern as follows:

1.      All the type II antitoxin does not have two operators, so title needs to reframe.

2.      They  have little description about the role of antitoxin as new antimicrobial candidate: line 266 -296 only.

3.      Figure number 3 is totally missing. One should crosscheck inclusion of all the figures after submission and before giving final approval for the same. Therefore, figure no 3 should be included.

Minor comments:

4.      From line number 33-41 references are missing.

5.      From line number 78-87 references are missing

Author Response

COMMENT 1> This is with reference to the review title “Dynamics-Based Regulatory Switches of Type II Antitoxins: Insights into New Antimicrobial Discovery by Ki-Young Lee and Bong-Jin Lee. They have discussed about the dynamic-based regulatory switched of type II antitoxin with specific reference to Phd-Doc/ CcdAB system having two operators. I have some major concern as follows:

All the type II antitoxin does not have two operators, so title needs to reframe.

RESPONSE 1> We appreciate the Reviewer’s kind evaluation of our manuscript. We have added descriptions about other subtypes of type II TA systems, which exhibit differences in the TA operon organization and the regulatory mechanism for TA transcription, in the chapter 2.2 “Regulation of TA transcription by antitoxin IDR” on pages 2-5 in the revised manuscript. Accordingly, we added relevant figures (B) and (C) to the Figure 1 on page 3. Thus, we did not reframe the title of manuscript.

COMMENT 2> They have little description about the role of antitoxin as new antimicrobial candidate: line 266 -296 only.

RESPONSE 2> To address the Reviewer’s comment, we have added discussions about our antibiotic strategies targeting antitoxin IDRs to the chapter 3 “Utilization of antitoxin IDR dynamics for new antibiotic discovery” on pages 7-10 in revised manuscript.

COMMENT 3> Figure number 3 is totally missing. One should crosscheck inclusion of all the figures after submission and before giving final approval for the same. Therefore, figure no 3 should be included.

RESPONSE 3> We confirmed that this figure is included on page 6 in the revised manuscript.

COMMENT 4> Minor comments:

From line number 33-41 references are missing.

From line number 78-87 references are missing

RESPONSE 4> As the Reviewer comments, we cited related references for these contents in the revised manuscript.

Reviewer 4 Report

The Perspectives manuscript titled “Dynamics-Based Regulatory Switches of Type II Antitoxins: Insights Into New Antimicrobial Discovery” extrapolates from other biochemical systems that rely on intrinsically disordered regions (IDRs) to propose a “dynamic allostery” of antitoxins underpinning the regulation of type II TA systems. The premise of the authors is that understanding this dynamic allostery will allow targeting type II TA systems as an antibacterial approach. 

Overall, there are some intriguing concepts and interpretations presented in the current work, although these are perhaps over generalized for dynamics in an isolated and closed state (which is very unlike the cellular environment). The connection to “New Antimicrobial Discovery” is a pervasive topic in the TA field, yet much of this work (including similar idea type papers) is not referenced. There are numerous opportunities to improve the manuscript by inclusion of appropriate citations. The writing itself is overall of high quality with clear presentation and connections of ideas. 

With some caveats in place, I feel this work contributes to the field of TA systems and, with the addition of appropriate citations, is in a publishable state. 

Specific questions (should the authors want to include information addressing them in the published version) about the concepts presented are:  

(a) that the dynamic ensembles suggested by the authors are occurring within a cell and are likely quite different than ex vivo models, such as those derived from NMR. Are the same dynamics expected in the crowded environment of a cell?  

(b) that the interaction with toxin is likely very fast (even as the toxin is being translated, as the critical interactions points are C-terminal antitoxin with N-terminal toxin for the canonically ordered operons) and early in the lifetime of these proteins, therefore much of the proposed IDR dynamics will be limited to antitoxin in excess of toxin 

(c) that the antitoxin is considered a hub bridging toxin neutralization and promoter repression; however, only a small fraction of antitoxin will be bound to the DNA promoter as there are only a few per cell, even when multiple operators on a given promoter are present (and there are many cases where only a single operator is present). How do antitoxins bound to toxin but not to DNA fit into the IDR models proposed? 

(d) it is very attractive to consider that the uncomplexed antitoxin could find a conformation that is less protease sensitive; however, I am not aware of any observation or measurement of such a result. This lack of experimental support tempers enthusiasm for this suggestion. 

Unfortunately, Figure 3 is missing from the copy for peer review; however, the information is interpretable without the figure. 

Figure 4 presents a nice summary of the suggested antibacterial approaches; of note, both strategies (I) and (ii) have published literature (unfortunately not cited here) demonstrating their utility, and strategies (iv) and (v) are completely hypothetical, despite many decades of research. This leaves strategy (iii), but it is unclear under what circumstances this would be useful – is it when TA systems have a demonstrated role in, for example, survival (such as Hig systems)? 

As mentioned above, there is an overall lack of citations for TA literature, while non-TA literature appears well cited; there are numerous examples that would be well-served to have citations, and some are listed below. 

Line 31, TA systems in pathogenic bacteria, and “attention as a novel antibiotic cellular target” 

Lines 75-75, association in picomolar to nanomolar ranges 

Line 80, “toxin neutralization induced by antitoxin binding has been proposed to employ” 

Line 85, “the crystal structures of TA complexes have revealed” 

Line 87, “IDRs alone are reluctant to crystalize” 

Line 122, “the low-to-high affinity transition has been thought as a key step” 

Line 123, “redundant toxin molecules in the cytoplasm” - where do these originate? Is the uncomplexed toxin present (outside of in theory)? 

Please cite the crystal structures in Figure 2 in addition to the PDB identifier 

Line 189, “has been accepted as a new paradigm” 

Line 196, “the common consensus is that” 

Line 198 “rapid proteolysis of the antitoxin is a critical step to liberate and activate the toxin.”  

Line 202, “influences cycling between the activation and inactivation states of toxins.” 

Line 204, “it appears that the antitoxin is protected from degradation once a stable TA complex is formed.” 

Line 207, “the formation of less stable non-cognate TA complexes leads to increased sensitivity to proteases, “ 

Lines 225-228 are highly speculative; “prevented the establishment of a reliable correlation” assumes a correlation, as does “disease-relevant specific conformations in the ensemble 

Author Response

COMMENT 1> The Perspectives manuscript titled “Dynamics-Based Regulatory Switches of Type II Antitoxins: Insights Into New Antimicrobial Discovery” extrapolates from other biochemical systems that rely on intrinsically disordered regions (IDRs) to propose a “dynamic allostery” of antitoxins underpinning the regulation of type II TA systems. The premise of the authors is that understanding this dynamic allostery will allow targeting type II TA systems as an antibacterial approach.  Overall, there are some intriguing concepts and interpretations presented in the current work, although these are perhaps over generalized for dynamics in an isolated and closed state (which is very unlike the cellular environment). The connection to “New Antimicrobial Discovery” is a pervasive topic in the TA field, yet much of this work (including similar idea type papers) is not referenced. There are numerous opportunities to improve the manuscript by inclusion of appropriate citations. The writing itself is overall of high quality with clear presentation and connections of ideas. 

RESPONSE 1> We appreciate the Reviewer’s kind evaluation of our manuscript. We carefully addressed the Reviewer’s points in the revised manuscript. Overall, we improved our manuscript mainly by adding more description and discussion about the regulatory mechanisms of type II TA systems and the antibiotic strategies to the chapter 2.2 on page 2-5 and the chapter 3 on page 7-10, respectively.

COMMENT 2> With some caveats in place, I feel this work contributes to the field of TA systems and, with the addition of appropriate citations, is in a publishable state. Specific questions (should the authors want to include information addressing them in the published version) about the concepts presented are:  

(a) that the dynamic ensembles suggested by the authors are occurring within a cell and are likely quite different than ex vivo models, such as those derived from NMR. Are the same dynamics expected in the crowded environment of a cell?  

RESPONSE 2> We thank the Reviewer for raising this valuable point. To address the Reviewer’s comment, we added the sentences “It should be considered that differentially populated dynamic states in the IDR ensemble can be influenced by the extremely crowded cellular environment that provides local high concentrations of biological macromolecules such as carbohydrates, nucleic acids, and proteins[52,53]. These crowding molecules could be inert or active components that form direct physical contacts with IDRs, potentially promoting either folding or unfolding of the IDR in a sequence-dependent manner. In particular, the crowded environment may cause considerable restrictions on the amount of free water and thus decreases volume availability of biomolecules, which can affect the conformational equilibrium of IDRs[54]. However, the crowding effect on IDRs of type II antitoxins remains unknown; for studying this conformational dynamics, in vitro biophysical techniques (e.g., NMR, SAXS, DLS) will be applicable in the presence of biologically inert crowders such as polyethylene glycol, Ficoll, and Dextran polymers [55-57]..” on page 7 in the revised manuscript.  

COMMENT 3> (b) that the interaction with toxin is likely very fast (even as the toxin is being translated, as the critical interactions points are C-terminal antitoxin with N-terminal toxin for the canonically ordered operons) and early in the lifetime of these proteins, therefore much of the proposed IDR dynamics will be limited to antitoxin in excess of toxin 

(c) that the antitoxin is considered a hub bridging toxin neutralization and promoter repression; however, only a small fraction of antitoxin will be bound to the DNA promoter as there are only a few per cell, even when multiple operators on a given promoter are present (and there are many cases where only a single operator is present). How do antitoxins bound to toxin but not to DNA fit into the IDR models proposed? 

(d) it is very attractive to consider that the uncomplexed antitoxin could find a conformation that is less protease sensitive; however, I am not aware of any observation or measurement of such a result. This lack of experimental support tempers enthusiasm for this suggestion. 

RESPONSE 3> We agree with the Reviewer’s comments. To improve our manuscript and address these Reviewer’s valuable comments [(b), (c), and (d)] regarding the basis of our antibiotic strategies targeting the IDR of antitoxins, we added more discussions to the chapter 3 “Utilization of antitoxin IDR dynamics for new antibiotic discovery” on page 7-10 in the revised manuscript.

COMMENT 4> Unfortunately, Figure 3 is missing from the copy for peer review; however, the information is interpretable without the figure. 

RESPONSE 4> We confirmed that the Figure 3 is included on page 6 in the revised manuscript.

COMMENT 5> Figure 4 presents a nice summary of the suggested antibacterial approaches; of note, both strategies (I) and (ii) have published literature (unfortunately not cited here) demonstrating their utility, and strategies (iv) and (v) are completely hypothetical, despite many decades of research. This leaves strategy (iii), but it is unclear under what circumstances this would be useful – is it when TA systems have a demonstrated role in, for example, survival (such as Hig systems)? 

RESPONSE 5> To address the Reviewer’s comment, we added more discussions about our five antibiotic strategies targeting the IDR of antitoxins in type II TA systems to the chapter 3 “Utilization of antitoxin IDR dynamics for new antibiotic discovery” on page 7-10 in the revised manuscript.  

COMMENT 6> As mentioned above, there is an overall lack of citations for TA literature, while non-TA literature appears well cited; there are numerous examples that would be well-served to have citations, and some are listed below.

Line 31, TA systems in pathogenic bacteria, and “attention as a novel antibiotic cellular target” 

Lines 75-75, association in picomolar to nanomolar ranges 

Line 80, “toxin neutralization induced by antitoxin binding has been proposed to employ” 

Line 85, “the crystal structures of TA complexes have revealed” 

Line 87, “IDRs alone are reluctant to crystalize” 

Line 122, “the low-to-high affinity transition has been thought as a key step” 

Line 123, “redundant toxin molecules in the cytoplasm” - where do these originate? Is the uncomplexed toxin present (outside of in theory)? 

Please cite the crystal structures in Figure 2 in addition to the PDB identifier 

Line 189, “has been accepted as a new paradigm” 

Line 196, “the common consensus is that” 

Line 198 “rapid proteolysis of the antitoxin is a critical step to liberate and activate the toxin.”  

Line 202, “influences cycling between the activation and inactivation states of toxins.” 

Line 204, “it appears that the antitoxin is protected from degradation once a stable TA complex is formed.” 

Line 207, “the formation of less stable non-cognate TA complexes leads to increased sensitivity to proteases, “ 

Lines 225-228 are highly speculative; “prevented the establishment of a reliable correlation” assumes a correlation, as does “disease-relevant specific conformations in the ensemble” 

RESPONSE 6> We thank the Reviewer for these kind comments. As the Reviewer pointed out, we removed or corrected inappropriate expressions in the revised manuscript.

Round 2

Reviewer 1 Report

I thank the authors because the manuscript is now improved but I have few minor comments:

- The figure 3 is missing

- Line 293: precise “type II TA systems”

- The legend of the figure 1 is misplaced in the text.

Author Response

COMMENT 1> I thank the authors because the manuscript is now improved but I have few minor comments:

- The figure 3 is missing

- Line 293: precise “type II TA systems”

- The legend of the figure 1 is misplaced in the text.

RESPONSE 1> We appreciate the Reviewer’s kind evaluation of our manuscript. 

As the Reviewer commented, we revised the manuscript. 

Reviewer 3 Report

This is with reference to the review title “Dynamics-Based Regulatory Switches of Type II Antitoxins: Insights into New Antimicrobial Discovery by Ki-Young Lee and Bong-Jin Lee. They have discussed about the dynamic-based regulatory switched of type II antitoxin. Revised version have addressed all the quries except the following :

1     1. Figure number 3 is  still missing in the manuscript. One should crosscheck inclusion of all the figures after re-submission and before giving final approval for the same. Therefore, figure no 3 should be included.